# NETosis in Parasitic Infections: A Puzzle That Remains Unsolved

**DOI:** 10.3390/ijms24108975

**Published:** 2023-05-19

**Authors:** Marwa Omar, Heba Abdelal

**Affiliations:** 1Department of Medical Parasitology, Faculty of Medicine, Zagazig University, Zagazig 44519, Egypt; 2LIS: Cross-National Data Center, Maison des Sciences Humaines, Esch-Belval, L-4366 Luxembourg, Luxembourg; omar@lisdatacenter.org

**Keywords:** extracellular traps, protozoa, innate immunity, NETosis, helminths, neutrophils, DNA

## Abstract

Neutrophils are the key players in the innate immune system, being weaponized with numerous strategies to eliminate pathogens. The production of extracellular traps is one of the effector mechanisms operated by neutrophils in a process called NETosis. Neutrophil extracellular traps (NETs) are complex webs of extracellular DNA studded with histones and cytoplasmic granular proteins. Since their first description in 2004, NETs have been widely investigated in different infectious processes. Bacteria, viruses, and fungi have been shown to induce the generation of NETs. Knowledge is only beginning to emerge about the participation of DNA webs in the host’s battle against parasitic infections. Referring to helminthic infections, we ought to look beyond the scope of confining the roles of NETs solely to parasitic ensnarement or immobilization. Hence, this review provides detailed insights into the less-explored activities of NETs against invading helminths. In addition, most of the studies that have addressed the implications of NETs in protozoan infections have chiefly focused on their protective side, either through trapping or killing. Challenging this belief, we propose several limitations regarding protozoan-NETs interaction. One of many is the duality in the functional responses of NETs, in which both the positive and pathological aspects seem to be closely intertwined.

## 1. Introduction

Parasitic helminths (metazoa) and protozoa are vastly diverse groups of eukaryotic organisms that cause various diseases in humans and animals. Parasitic infections pose a major public health concern in developing countries, mainly in tropical and subtropical communities [1]. Parasites can cause persistent infection due to their ability to resist immune-mediated expulsion by modulating the host’s immune response [2,3,4]. The intricate interaction between the parasite and the host is critically important when understanding the pathophysiology of the diseases they cause [5].

Innate immunity is an ancient form of the host’s defensive mechanisms against invasive pathogens. This type of immunity has evolved to protect the host against different infectious agents, including bacteria, fungi, viruses, and parasites [6]. The components of the innate machinery include physical and anatomical barriers, along with humoral and cellular factors [7,8]. Neutrophils, monocytes, and macrophages are among the professional phagocytic cells participating in both phagocytosis and inflammatory processes [9,10].

The generation of extracellular traps (ETs) has been recognized as a novel mechanism of the innate immune response against infections. These traps are expelled to facilitate the immobilization and killing of invading microorganisms in the extracellular environment [11]. ETs can be generated by several leukocytes, including neutrophils, eosinophils, basophils, and mast cells [12]. In addition, activated T cells, B cells, and monocytes are likely to release mitochondrial DNA (mtDNA), forming extracellular web-like structures. Regardless of the type of cells from which they originate, the traps share a common feature: they consist of nuclear or mtDNA as a backbone, with embedded antimicrobial peptides, histones, and cell-specific proteases [13].

## 2. Neutrophils and Neutrophil Extracellular Traps (NETs)

### 2.1. Neutrophils

Neutrophils, or poly-morphonuclear (PMN) leukocytes, are the most abundant cell type circulating in human blood. They are considered the foot soldiers of the immune system and the first comers to the sites of infection or inflammation [14,15]. Neutrophils play a pivotal role in developing innate and adaptive immune responses [16]. They employ different strategies to combat pathogens, including phagocytosis [17,18], degranulation [19,20], and the formation of neutrophil extracellular traps (NETs) [21].

### 2.2. NETs and NETosis

The original description of NET production was first introduced by Brinkmann et al. (2004) as a new defensive mechanism consisting of the ejection of intracellular material in the form of web-like elements into the surrounding extracellular medium. The authors discovered that when stimulated with phorbol myristate acetate (PMA), lipopolysaccharide (LPS), and interleukin 8 (IL-8), neutrophils release nuclear DNA fibers after exposure to Gram-positive or Gram-negative bacteria [21]. NETs are composed of a backbone of DNA decorated with histones and laced with several cytoplasmic antimicrobial granular proteins [21,22,23]. The granular effector molecules include myeloperoxidase (MPO), neutrophil elastase (NE), lactoferrin, cathepsins, pentraxin, gelatinase, peptidoglycan recognition protein (PRP), calprotectin, bacterial permeability-increasing protein (BPIP), and other leukocyte peptides [24,25,26]. The fibrous DNA strands in the NETs are punctuated with globular protein domains in large aggregates ranging from ~25 nm up to 50 nm in size. The chromatin constitutes the backbone of these structures, as NETs can be degraded by DNases, but not by proteases [27].

The whole process of NET production is called NETosis, a phenomenon first described in the work of Takei et al. (1996) [28], based on the term ‘NET’ [22] and the Greek suffix ‘-osis’ [29]. It is a novel form of cell death that is distinct from autophagy or necrosis, as it is independent of the caspase pathway, and there is no phosphatidylserine exposition on the cell surface [22]. Furthermore, NETosis is not associated with DNA fragmentation, as is the case in apoptosis, but rather with the release of NETs after the disruption of the plasma membrane and nuclear envelope. The sticky DNA-rich networks not only entrap invading pathogens, but eventually kill them [30].

Various reports have defined NETosis as an NADPH oxidase (NOX)-dependent mechanism [21,31]. Under stimulation, neutrophils generate reactive oxygen species (ROS) by NADPH oxidase-2 (NOX-2) [32]. ROS modulate the granular enzyme MPO, and both are required for the release of NE from granules and its translocation to the nucleus [33,34]. In the nucleus, NE contributes to chromatin decondensation by the proteolysis of histones. Such events result in the nuclear extrusion of DNA and NET production [22,32]. Yet, other studies have described a (NOX)-independent NET formation process, which is facilitated through calcium influx and mitochondrial ROS production [35,36]. Hence, this type of NET expulsion can be called “mitochondria-dependent NETosis” [37].

### 2.3. Mechanisms of NET Formation

There are at least three different mechanisms by which NETs are formed: classical or suicidal NETosis, the noncanonical pathway, and vital NETosis. The lytic or suicidal type usually occurs slowly (2−4 h) and involves the rupture of the neutrophil plasma membrane. On the other hand, nonlytic or vital NETosis occurs rapidly (within minutes) and does not involve disruption to the plasma membrane or cell lysis [38]. During vital NETosis, neutrophils are still capable of functions such as migration, degranulation, and phagocytosis while casting their sticky filaments [39]. The noncanonical pathway is a new pathway of lethal NETosis that has recently been described. This sort suppresses bacterial residence in the neutrophil cytosol and prevents in vivo microbial dissemination [40].

### 2.4. Microbial Triggers of NETosis

Multiple stimuli promote the release of NETs. These include the interaction of neutrophils with other immune cells (platelets) after activation with cytokines (IL-8) that help to entrap and eliminate pathogens [41]. Additionally, neutrophils induce vital NETosis when exposed either in vitro or in vivo to whole microorganisms or their proteins [39]. The microbicidal effects of NETs have been confirmed in several human and animal models with bacterial [21], viral [42], and fungal infections [43].

NETs entrap many types of bacterial pathogens to prevent their spread. A variety of Gram-positive and Gram-negative bacteria have been shown to induce NET expulsion. Infection with the swine pathogen *Streptococcus suis* leads to the signaling of NET formation pathways in a NOX-dependent approach [44]. Additionally, the Gram-negative bacterium *Klebsiella pneumoniae* proved to be a good inducer of NETosis in a mouse lung infection model [45]. Yet, some bacteria have developed strategies to resist their capture or elimination by the NET structures. *Staphylococcus aureus* produces different enzymes to interfere with the antimicrobial properties of NETs, such as a nuclease (Nuc), a DNA binding protein, and the extracellular adherence protein (Eap) [46]. These enzymes facilitate the escape from the NET filaments, delay bacteria clearance, and increase the mortality caused by the infection [47]. Besides the reported role of bacteria in triggering NETosis, increasing evidence indicates that viruses can also promote NET formation, which may either promote or prevent viral-induced pathology [48]. In the case of Human Immunodeficiency Virus-1 (HIV-1), NETs promote pathogen clearance through the actions of histones and MPO [49]. Recently, Veras et al. [50] demonstrated an increase in the quantity of the NET components (DNA-MPO complex) in the plasma, tracheal aspirate, and autopsy lung tissues of COVID-19 patients. The study highlighted a possible detrimental role of NETs in the pathophysiology of COVID-19.

Neutrophils and lymphocytes are closely related to the pathophysiology of several inflammatory disorders. The neutrophil-to-lymphocyte ratio (NLR) is a biomarker that conjugates two arms of the immune system: the innate immune response, mainly due to neutrophils, and adaptive immunity, promoted by lymphocytes. This ratio is characterized by an increase in neutrophils and a decline in lymphocytes [51]. In their research, Regolo et al. (2022) tried to assess the prognostic value of (NLR) as a predictor of the outcome of COVID-19 patients. According to them, patients with higher NLR showed a higher risk of intra-hospital mortality and disease progression [52].

The filamentous fungus *Candida albicans* (*C. albicans*) is the most widely discussed fungal pathogen in the field of NET release. Following the interaction of neutrophils with *C.albicans,* the granular protein (NE) is released into the cytoplasm to initiate the production of NETs, which have been shown to kill both yeast and hyphal forms of the fungus [43]. Yet, *C. albicans* has its unique way of resisting NETs. It potentially modulates the formation of NETs by arresting their proteinaceous components, including elastase, MPO, and histones. Additionally, adhesins on the fungal surface can adsorb the NETs proteins and increase the pathogen’s potency in host tissue destruction [53].

## 3. NETosis in Parasitic Infections

The functional responses of neutrophils to parasitic infections continue to be uncovered. It has been suggested that the size of the stimulating particle is one of the factors driving the decision of neutrophils to generate NETs instead of phagocytosis. While large particles, such as parasites, can induce the formation of NETs, small particles, such as bacteria, yeast, or viruses, can be eliminated by phagocytosis [54,55]. Knowledge is not yet abundant about the involvement of NETs in the host’s innate response against metazoan and protozoan parasites [5].

In the current study, we have presented NETs as informative demonstrations to provide clues to their potential actions in the host’s battle against several parasitic species (Figure 1).

### 3.1. NETosis and Metazoan Infections

So far, few data are available about metazoan-induced NETosis [56]. Herein, we aspire to outline the possible functions employed by NETs during different helminthic infections, which can be broadly grouped into three categories (Figure 2):Trapping without killing;Blocking the development;Direct killing (larvicidal) effects.

#### 3.1.1. Trapping without Killing

Different studies have demonstrated the ability of neutrophils to cast large amounts of NETs that capture but do not kill the parasite. McCoy et al. (2017) demonstrated the in vitro production of NET aggregates in response to *Brugia malayi* (*B.malayi*) microfilariae. In their research, they found that both human neutrophils and monocytes were able to kill the microfilariae in vitro, whereas NETs only facilitated the attachment of leukocytes to Malayan microfilariae without killing them [57]. Additionally, *Dirofilaria immitis* (*D.immitis*) infectious third-stage larvae (L3) and microfilariae are potent inducers of canine NETosis. Quantitative analyses showed that both larval stages triggered the rapid and pronounced formation of canine NETs in a time-dependent manner. Yet, neither D. immitis microfilariae nor the infective larvae were killed by the NET traps [58]. *Strongyloides stercoralis* (*S. stercoralis*) large and highly motile larvae have also triggered the release of extracellular DNA traps by human neutrophils and macrophages. The in vitro assays performed by Bonne-Année et al. (2014) proved that despite the induction of NETs by human neutrophils, the NET-based clots trapped but did not kill *S. stercoralis* larvae, as the live larvae were seen moving within these clots. Nonetheless, the extrusion of fine DNA threads was necessary for larval killing by human macrophages and neutrophils (Figure 2A) [59].

There are three distinctive forms of NETs that have been described so far: “diffuse” NETs (diffNETs), “spread” NETs (sprNETs), and “aggregated” NETs (aggNETs). The diffNETs are composed of a decondensed extracellular chromatin complex decorated with compact antimicrobial proteins, while sprNETs consist of elongated, smooth structures in the form of a decondensed chromatin web and antimicrobial proteins. The aggregated forms (aggNETs) are composed of clusters of NET-like threads with a “ball of yarn” appearance [58,60]. Such phenotypes were expressed by bovine neutrophils upon contact with *Haemonchus contortus* (*H.contortus*) L3. These DNA webs contributed to the time-dependent ensnarement of the larvae but did not participate in parasite killing [60]. In their experiment, the authors observed that within the definitive host, NETs could limit the establishment of *H. contortus* infection by preventing active larval migration into the site of infection. The entrapped immobilized larvae were then susceptible to other immunocompetent cells, such as monocytes and macrophages, exhibiting larvicidal effects.

Recently, trematode-mediated NETosis was reported in the *Fasciola* species. In their study, Guo et al. (2020) indicated that *Fasciola gigantica* (*F. gigantica*) newly excysted juveniles (NEJs) triggered the release of NETs following their incubation with water buffalo neutrophils. The sticky threads only entrapped the motile flukes and hampered their migration. However, they did not influence the fluke viability or integrity. The authors suggested that fluke-triggered NET formation might serve as a mechanism to hinder large parasitic movement, thereby facilitating immune cells to kill the large fluke [56]. *Fasciola hepatica* (*F. hepatica*) eggs, metacercariae, and NEJs were identified as weak inducers of bovine NETosis. In their experiment, Peixoto and co-authors [61] recorded that only a few bovine (PMN) extruded fine and short sprNETs after exposure to different stages of the *F. hepatica* parasite. According to them, bovine sprNETs do not kill these parasitic stages, yet might promote their killing by other leukocytes recruited into F. hepatica parasitized tissues. In line with this, the co-culture of human neutrophils with viable *Schistosoma japonicum* (*S. japonicum*) eggs resulted in the release of extracellular DNA fibers, which were attached to the surface of the eggs. Despite the containment and immobilization, the eggs were not killed by the NETs, as the intact nuclei of the schistosome embryos within the eggs were visible even after the eggs were trapped (Figure 2A) [62].

#### 3.1.2. Blocking the Development

In 2022, a study showed that the extracellular vesicles (EVs) derived from the liver tissues of mice experimentally infected with *S. japonicum* induced NET production by targeting neural Wiskott–Aldrich syndrome protein (WASL). The loss of the WASL gene accelerated neutrophil-induced NET formation, which prevented the development of the *S. japonicum* worms. This blockage induced a significant reduction in egg deposition, thereby attenuating the pathological progression of the infection. The expulsion of NETs caused a marked elevation in the expression of the inflammatory C-C chemokine ligand 2 (CCL2) in macrophages, which were recruited to further block the development of *S. japonicum* and ameliorate host fibrosis (Figure 2B) [63].

#### 3.1.3. Direct Killing (Larvicidal) Effects

In the murine hookworm *Nippostrongylus brasiliensis* (*Nb*), neutrophil extracellular webs were released around the infective skin penetrating (L3). Bouchery et al. (2020) proved that these NETs alone could directly kill hookworm larvae. After using the in vitro co-culture assay of neutrophils and parasitic larvae, the authors confirmed that NETosis occurs in response to both live and dead *Nb* larvae. However, the NETs were only observed in cultures of dead larvae, suggesting that live larvae secrete a DNase capable of degrading the DNA backbone of the NET threads. The addition of an anti-serum to block the (DNase II) enzyme resulted in the impairment of parasite survival and enhancement of larval killing, confirming the direct larvicidal effects of NETs during murine hookworm infection [64]. The human hookworm *Necator americanus* (*N. americanus*) was also evinced to induce NETosis in vitro. The DNA webs exhibited both trapping and direct killing properties against the infective (L3) larvae by inducing cuticle damage (Figure 2C), resulting in increased permeability to the DNA binding dye Sytox-Green. Temperature activation enhanced the larval secretion of the DNase-II enzyme, which was capable of degrading the NET strands to escape trapping and cuticle damage [65].

#### 3.1.4. Helminths Evading NET-Mediated Killing

As mentioned above, NETs appear to be serving numerous functions in response to those multicellular macroparasites that might be too large to be easily phagocytosed. However, it seems evident that several helminths are endowed with diverse strategies to overcome the NETs killing machinery [66]. To date, reports on parasitic factors with the capacity to inhibit, degrade, and evade the antimicrobial activities of NETs are still lacking [67]. The expression and secretion of virulence factors are among the mechanisms employed by metazoa to escape the innate immune response. Some helminths respond to NETs by producing excretory-secretory (ES) antigens, which are potent modulators of the immune system [68]. These molecules are released at the interface between the parasite and the immune cells to ensure sustained parasitic maintenance and long-term survival within the infected host [69]. Both nematodes (roundworms) and platyhelminths (flatworms) release ES molecules that potentially inhibit neutrophil-mediated NET production, thereby evading parasitic containment and elimination. Ríos-López et al. (2022) proved that *Trichinella spiralis* (*T. spiralis*) ES antigens inhibited the release of NETs from neutrophils without affecting their phagocytic or chemotactic activities [70]. This observation was supported by another report that identified a novel NET inhibitory factor of the tapeworm *Mesocestoides corti* (*M. corti*). The parasitic ES molecules (termed parasitic ligands (PLs)) inhibited the in vitro generation of NETs in response to H_2_O_2_. Moreover, the treatment of mice with PLs improved their survival in septic peritonitis by preventing NETosis, resulting in enhanced bacterial clearance [71].

Serine proteinase inhibitors were among the reported factors contributing to the development and survival of the parasitic nematodes within their hosts [72]. In their investigation, Milstone et al. (2000) identified the first serine proteinase inhibitor isolated from a major hookworm, *Ancylostoma ceylanicum* (*A. ceylanicum*). Kunitz-type serine protease inhibitor-1 (AceKI-1) played a significant role in parasitic protection inside the intestinal tract by blocking the activities of neutrophil elastase, pancreatic elastase, and trypsin [73]. Another serine proteinase inhibitor was isolated from the adult stage of the human hookworm *Ancylostoma duodenale* (*A.duodenale*). The novel serine protease inhibitor, named AduTIL-1, had two trypsin inhibitor-like (TIL cysteine-rich) domains, with inhibitory effects on human neutrophil elastase and digestive pancreatic trypsin, hence protecting the worm from the damage of the related digestive enzymes within the host intestine [74].

#### 3.1.5. NETs Acting as a Cloaking Device

Various reports have confirmed the trapping actions of the NETs being exploited against different metazoan species. During such infections, the DNA strands capture the worms to facilitate recruitment, attachment, and killing by various immune cells, such as neutrophils, monocytes, and macrophages [57,58,59]. Yet, a rather intriguing function of the NETs was observed during human infection with the filarial nematode *Onchocerca volvulus* (*O.volvulus*), the causative agent of onchocerciasis (river blindness). The worm harbors an intracellular bacterial symbiont, *Wolbachia*. *Wolbachia* are maternally transmitted, Gram-negative bacteria that form a spectrum of endosymbiotic relationships, from parasitism to obligatory mutualism, in a wide range of arthropods and nematodes. The host inflammatory response to the dying *O.volvulus* microfilariae and the subsequent release of their bacterial content is thought to form the basis of onchocercal dermatitis and ocular keratitis [75]. In their research, Tamarozzi and colleagues [76] demonstrated the extrusion of extracellular threads around *Wolbachia*-containing adult worms in nodules excised from *O.volvulus*-infected untreated patients. The presence of NETs served as a “cloaking device” or a physical barrier to decrease the penetrance of more damaging immune cells, such as eosinophils, toward the cuticle surface of adult *Onchocerca*. This enveloping effect of NETs might help prevent the dissemination of *Wolbachia*, thus limiting the inflammatory damage and immunopathological sequelae induced by endosymbiotic bacteria (Figure 3) [76].

Table 1 lists the evasive approaches evolved by different helminths to overcome the actions of NETs and the mechanisms thereof.

### 3.2. NETosis and Protozoan Infections

When compared to the studies conducted on other microbial pathogens, the role of NET production in answer to protozoans still seems to be underrepresented. The process of NETosis has been identified during infections with *Eimeria, Toxoplasma, Trypanosoma, Plasmodium,* and *Leishmania* parasites [77,78,79,80,81,82]. Yet, until now, less has been known about the multitude of functions that the DNA networks might play during the invasion of host tissues by different protozoa. The positive roles of NETs against protozoa are balanced by their negative impacts on the health of the infected hosts [83]. The associations of NETs with inflammatory reactions observed in established protozoan infections raise the possibility of their participation in pathology instead of protection [84]. The protective and deleterious aspects of NETosis encountered during the progression and pathogenesis of different protozoan infections are explained below.

#### 3.2.1. The Protective Roles of NETs during Protozoan Infections

##### Microbicidal Effects of NETs

Despite the increasing number of studies that have explored the functions of neutrophils in the early immune response to protozoa, few data are available on the precise role of NETosis in the outcome of protozoan infections. NETs have been recognized as a neutrophil-killing mechanism for multiple protozoans [85]. One of the earliest investigations conducted to prove the antimicrobial potential of NET-related molecules mainly involved the *Leishmania* species. Guimarães-Costa and co-authors [86] tested the extrusion of NETs by human neutrophils upon their interaction with *Leishmania amazonensis (L. amazonensis)* in vitro. According to their findings, the NET-mediated killing of promastigotes depended on the toxic properties of the histone component. This lethal effect was inhibited by adding anti-histone antibodies to the neutrophil–promastigote interaction medium [86]. Additionally, both of the NET-derived structures (elastase and histone) further participated in controlling the parasitic burden in *Leishmania braziliensis (L. braziliensis)* infections. A positive correlation was observed between the parasite load and the number of NETs in the active lesions of patients with American tegumentary leishmaniasis (ATL). The dynamic process of NET production surrounding the extracellular amastigotes resulted in protozoan killing and the impairment of *Leishmania* dissemination [87].

One of the NET-associated toxins is the MPO enzyme, for which the role of eliminating *Entamoeba histolytica* (*E.histolytica)* infection was recently investigated. In vitro cultures of amoebae and rodent neutrophils (from mice and hamsters) induced the expulsion of NETs, with the concentrations being higher in the mice than in the hamsters, a factor that caused a significant reduction in amoebic viability in the culture of amoebae and mouse neutrophils. The anti-parasitic activity of MPO was confirmed by using the 4-amino-benzoic acid hydrazide (ABAH) inhibitor, which induced a marked increase in trophozoite viability [88]. Recently, Ramírez-Ledesma et al. (2022) examined the in vitro interaction of *Trichomonas vaginalis (T.vaginalis)* with human neutrophils. In their investigation, the growth of trichomonads decreased by at least 60% due to the entrapment effects of NETs, which interfered with protozoan integrity and survival [89].

##### Prevention of Host Cell Invasion

The previous findings correspond well with the defensive roles of NETs against the pathogenic protozoan *Toxoplasma gondii* (*T. gondii*). Following in vitro exposure to the tachyzoite stages, the free DNA webs released from human, mouse, and cattle neutrophils promoted the extracellular killing of *Toxoplasma*, thereby interfering with the parasite’s ability to invade target host cells [78,90]. The sticky NET strands have also contributed to blocking the life cycle at the onset of infections regarding the apicomplexan protozoa *Eimeria bovis (E.bovis)* and *Cryptosporidium parvum (C. parvum)*. After incubating bovine neutrophils within the sporozoite stages of both parasites, the generated fibrous traps induced the immobilization of the sporozoites, a mechanism that subsequently hampered host cell invasion and parasitic multiplication [77,85].

#### 3.2.2. Achilles’ heel of NETs during Protozoan Infections

Despite the widely reported protective effects of NETs against different protozoa, some drawbacks were noticed for these effects. It appears feasible to refer to these as the Achilles’ heel of NETs, which can be classified into the following groups (Figure 4):NETs lacking the killing capacity;Evading NET-mediated killing;NETosis-induced pathology.

##### NETs Lacking the Killing Capacity

*C. parvum*-induced NET formation has been investigated in both human and bovine neutrophils. Despite the significant entrapment and immobilization of sporozoites in the NET filaments, evidence of a lack of NET-mediated killing was revealed after (Dnase-I) treatment. The inhibition of the NET-induced invasion was reversed by the (Dnase-I) enzyme, which verified that the *C. parvum* parasite was not killed by NETs, as was postulated for different bacterial pathogens [91]. Additionally, the in vitro exposure of caprine (PMN) to the parasitic stages (sporozoites and oocysts) of the goat protozoa *Eimeria arloingi (E.arloingi)* and *Eimeria ninakohlyakimovae (E. ninakohlyakimovae)* triggered the formation of NETs, which firmly entrapped the parasites. Yet, the trypan blue exclusion test confirmed that the NET traps did not interfere with the viability of sporozoites [92,93].

The lethal impact of NETs on the tachyzoite stage was widely explored in the apicomplexan parasites *Besnoitia Besnoiti (B. besnoiti)*, *Neospora caninum (N. caninum),* and *T.gondii.* The co-cultivation of bovine (PMN) with the obligatory intracellular protozoa *B. besnoiti* and *N. caninum* strongly induced the production of NETs, which entrapped tachyzoites and prevented their subsequent invasion of host cells. Nevertheless, after an incubation period of 3 h, the NET filaments failed to exhibit any tachyzoite-killing activity (Figure 4A) [94,95]. Similarly, the NET meshwork released by sheep (PMN) against the *T.gondii* parasite in vitro only ensnared the tachyzoites without killing them [90].

The trypomastigote stages of different *Trypanosoma* species were described as potent inducers of NET production. The motile trypomastigotes of *Trypanosoma brucei* (*T. b. brucei*) triggered the release of different NET phenotypes in bovine PMN, such as (*diff*NETs), (*agg*NETs), and (*spr*NETs). Such traps slightly decreased the motility of trypomastigotes without affecting their viability (Figure 4A) [96]. Recently, Wei et al. (2021) investigated the effect of NETs on the viability and motility of *Trypanosoma evansi (T. evansi).* Their results showed that the NETs extruded by mouse neutrophils did not affect the viability of the protozoan, despite reducing its motility [97]. Similar observations were reported for the Y strain of the *T. cruzi* parasite. In their research, Sousa-Rocha and coworkers [98] proved that the NETs generated by human neutrophils could not kill *T. cruzi*, but interfered with the pathogenicity and infectivity of the parasite. The relevant euglenozoan parasite *Leishmania mexicana (L. mexicana)* elicits NET formation in murine models. The in vitro exposure of mouse neutrophils to the metacyclic promastigotes of *L. mexicana* induced the generation of NETs, which trapped the parasite without markedly impairing its survival [99].

During *E. histolytica* infection, Ávila et al. (2016) observed that human NETs were unable to kill the trophozoite stages in vitro [100]. Likewise, the process of NETosis did not contribute to the neutrophil-mediated killing of *T*. *vaginalis* protozoan [101].

##### Evading NET-Mediated Killing

The *Leishmania* protozoan was recognized as the best-described example of parasites escaping the antimicrobial activities of NETs. Gabriel and colleagues [102] investigated the evading potential exhibited by *Leishmania donovani (L. donovani)* promastigotes against NETs generated by human neutrophils. Their study showed that the dense glycocalyx, lipophosphoglycan (LPG), that covers the promastigote surface acted as a physical barrier to protect the parasite from the different microbicidal molecules associated with NET structures [102]. Additionally, the surface glycolipid LPG conferred *Leishmania major (L. major)* promastigotes with the ability to resist the toxicity of NET histones [103]. Another evasive strategy was employed by *Leishmania infantum (L. infantum).* The cultivation of the promastigote stages in a low-phosphate (LP) medium resulted in the greater expression of the parasitic 3′-nucleotidase/nuclease (3′NT/NU) enzyme, which allowed the parasite to cleave the released NET traps and thereby escape their toxic effects (Figure 4B) [104]. Recently, Zhang et al. (2021) identified two TatD DNases for *T. brucei* and *T. evansi* parasites. These enzymes were localized in the cytoplasm and flagella of both protozoa and possessed efficient DNA hydrolytic activities, which allowed *Trypanosoma* to disrupt the filamentous elements of the NETs and, hence, evade killing (Figure 4B) [105].

##### NETosis-Induced Pathology

The role of NETosis in either the protection against *Plasmodium* infection or tissue damage resulting in the pathogenesis of severe malaria remains controversial. The accumulated data implicate NET molecules in the pathophysiology of human malaria. Feintuch et al. (2016) demonstrated elevated plasma concentrations of the neutrophil granular proteins NE, MPO, and proteinase 3 (PRTN 3) in cerebral malaria (CM)-positive Malawian children [106]. A positive correlation was also confirmed between the NET counts and parasite biomass in human falciparum malaria, in which the number of NETs increased in proportion to the severity of the disease [81]. Furthermore, Knackstedt and colleagues [107] analyzed the plasma samples from patients infected with *Plasmodium falciparum (P. falciparum)* in Gabon. The NETs were significantly enriched in severe versus uncomplicated malaria. The authors have further examined retinal tissue from fatal pediatric cases in which the patients died of CM, and NETosis was detected exclusively in the retinopathy-positive CM cases [107].

Sercundes and co-authors [108] reported the first implication of NETs in the development of malaria-associated respiratory distress in *Plasmodium berghei* ANKA (PbA)-infected mice. In their investigation, the inhibition of NET formation using recombinant human DNase or elastase inhibitors significantly reduced lung lesions and improved mouse survival [108].

Different mechanisms have been introduced to explain the NET-mediated complications in severe *P. falciparum* malaria. Boeltz et al. (2017) proposed that the intravascular formation of NETs contributes to the vasculopathy driving complicated malaria. According to them, NETs promote fibrin deposition and platelet adhesion. Therefore, the activation of the coagulation cascade and recruitment of platelets to intravascular NETs lead to microvascular occlusion and the disruption of the blood–brain barrier (BBB) (Figure 4C) [109]. By using the *Plasmodium chabaudi (P. chabaudi*) mouse model, Knackstedt and colleagues [107] defined another mechanism of NET-mediated inflammation. They identified NETs as a source of immunostimulatory molecules (alarmins) that activate emergency granulopoiesis by stimulating the release of granulocyte colony-stimulating factor (GCSF). This activation induced neutrophil trafficking into the livers of parasitized mice, with subsequent tissue destruction due to neutrophil cytotoxic molecules [107].

The hypothesis that excess NET production might contribute to parasite-derived pathology was additionally emphasized in *Leishmania* infection. In their retrospective study, Gardinassi et al. (2017) evaluated gene expression data from whole blood samples of visceral leishmaniasis (VL) patients in Brazil. The authors demonstrated a significant association between the transcriptional profile of VL patients and NET-associated proteins. Based on their findings, the release of NETs during the chronic stage induces a systemic inflammatory response, contributing to disease pathogenicity [110]. The inflammatory nature of NETs has also been extended to include *E. histolytica*. NETs (mtDNA) induced by trophozoite stage promoted the inflammation and tissue damage caused by *E. histolytica* [111]. Regarding the effects of NETs on infected endothelial cells, Conejeros et al. (2019) demonstrated damage and cytotoxicity induced by the major NET component histone 2A (H2A) in *B. besnoiti*-infected bovine endothelial cells [112].

Previous data reflect conflicting reports on the exact roles of NETs during infection by different protozoa. Table 2 summarizes this paradoxical nature of NETs, which can be referred to as the Jekyll and Hyde effect.

## 4. Conclusions

Almost 20 years ago, a previously unanticipated antimicrobial function was described for neutrophils, which entails the expulsion of web-like DNA filaments into the extracellular milieu to arrest and kill invading pathogens. While much has been revealed about the role of neutrophil extracellular traps (NETs) against viruses and bacteria, research is slowly progressing toward the NET-derived impact on parasitic infections. As reviewed here, several parasites induce NETosis both in vivo and in vitro. It also appears undeniable that some species have evolved different evasive mechanisms to either block NETs formation or degrade them once formed, raising questions about the immunological factors that drive the actions of NETs toward particular parasites. In *L. donovani* infection, dense glycocalyx (LPG) acts as a physical barrier to protect the parasite against the toxic activities of NETs [102]. In contrast, during human infection from *O.volvulus,* the NET chains serve as a barrier or cloaking device that envelops the parasite and alters the course of the disease [76].

This review shows how the extracellular networks switch roles between defense and damage. The determination of the signals and immune mediators behind this regulation might contribute to unfolding effective control strategies in cases where adverse effects are conferred by the production of NETs.

## Figures and Tables

**Figure 1 ijms-24-08975-f001:**
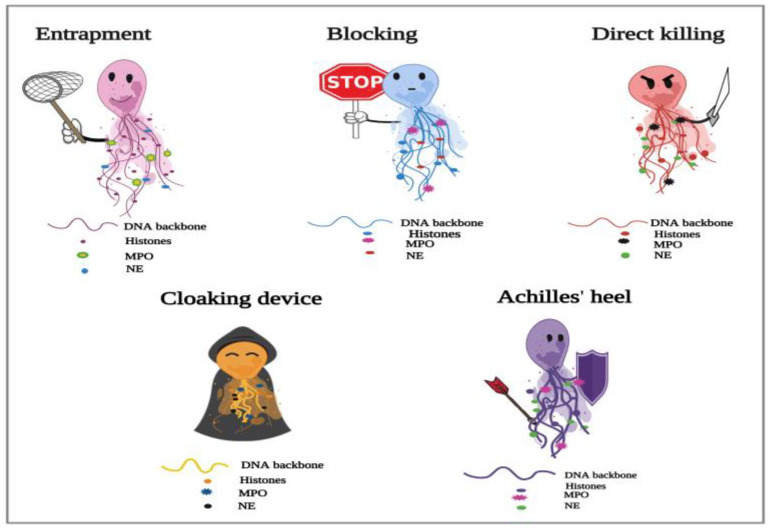
Proposed depiction of the functional responses of NETs to parasitic infections. In the process of NETosis, neutrophils release complex DNA webs, in which actions extend from parasitic entrapment to blocking the worm development and effector killing. Further, NETs could serve as a “cloaking device” to control the spread of infection. We also have created the “Achilles’ heel of NETs” to frame the several reported limitations in their actions in answer to different parasites.

**Figure 2 ijms-24-08975-f002:**
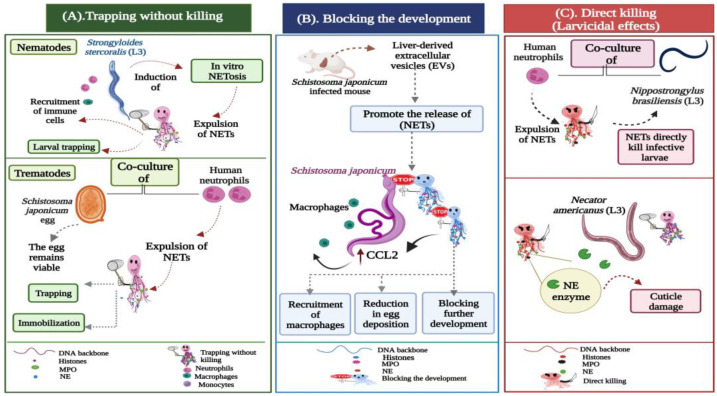
Potential actions operated by NETs during metazoan infections. (**A**) Trapping without killing: *Strongyloides stercoralis* large and motile larvae are potent inducers of in vitro NETosis. During the infection, the DNA threads ensnare larvae to enhance their killing by the recruited leukocytes, neutrophils, and macrophages. In trematode-triggered NETosis, the NET filaments capture the egg stages of *Schistosoma japonicum.* NETs hinder the mobility of the eggs, which still remain viable and intact within the fibrous traps. (**B**) Blocking the development: Extracellular vesicles (EVs) isolated from the liver of *Schistosoma japonicum-*infected mice initiate the expulsion of NETs, which block worm development, resulting in a significant reduction in egg deposition and associated fibrosis. The extruded NETs also promote chemokine ligand 2 (CCL2) expression in macrophages, which adhere to the surface of *Schistosoma japonicum* worms, inhibiting their further development. (**C**) Direct larvicidal effects: The direct toxicity of neutrophils is distinctly exhibited against the infective stages of hookworms. The skin-penetrating third-stage larvae (L3) of *Nippostrongylus brasiliensis* and *Necator americanus* become mechanically trapped by NETs, which directly participate in the larval killing using the neutrophil elastase (NE) enzyme, which induces cuticle damage to the hookworm larvae.

**Figure 3 ijms-24-08975-f003:**
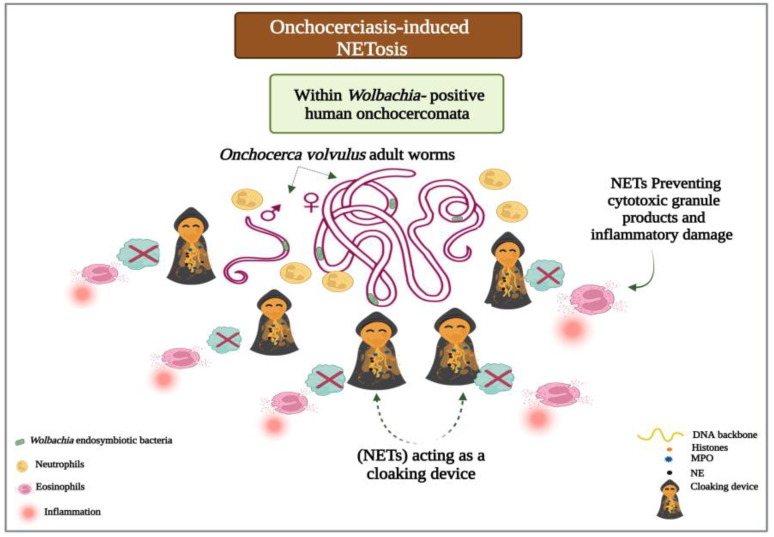
Onchocerciasis-induced NETosis: NETs serving as a cloaking device. In human onchocerciasis, the process of NETosis is triggered by *Wolbachia* endobacteria. Neutrophils cast sticky filaments, which envelop viable worms at the site of the infection to entrap and limit bacterial dissemination. Acting as a physical barrier, NETs surround the adult *Onchocerca*, blocking the access and recruitment of eosinophils close to the cuticle surface of the filarial nematode to protect host tissues from *Wolbachia*-induced inflammation and immunopathology.

**Figure 4 ijms-24-08975-f004:**
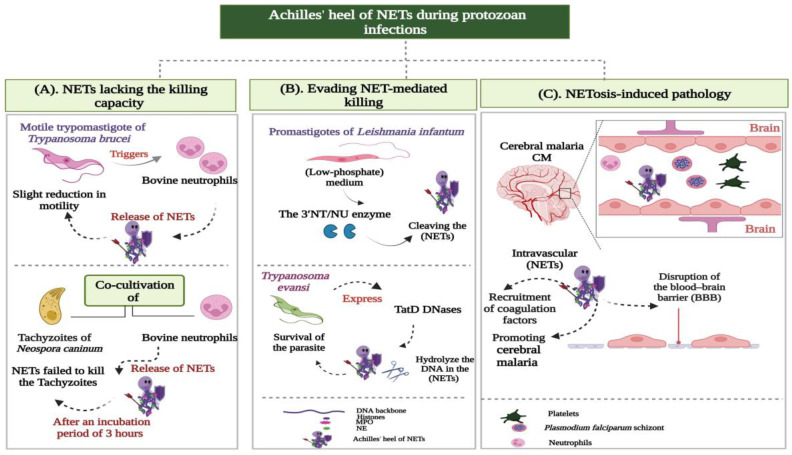
Achilles*’* heel of NETs during protozoan infections. (**A**) NETs lacking the killing capacity: Despite the defensive roles of NETs against different protozoa, an evident lack of NETs-dependent killing is displayed against different parasites. The euglenozoan protozoan *Trypanosoma brucei* triggers the release of different NET phenotypes in bovine neutrophils. These traps interfere with the motility of trypanosomes without affecting their viability. NETs also ensnare the tachyzoites of *Neospora caninum* to prevent them from invading target host tissues. Yet, they fail to elicit any lethal activities against them. (**B**) Evading NET-mediated killing: *Leishmania* species constitute the chief example of parasites evading the NETs killing machinery. The parasitic enzyme 3′-nucleotidase/nuclease (3′NT/NU) cleaves the NET webs around *Leishmania infantum*, allowing the protozoan to escape their toxicity. The survival of *Trypanosoma evansi* parasite is associated with its active secretion of the trypanosomatid (TatD DNases), which could efficiently hydrolyze the NET-DNA to facilitate trypanosomal evasion. (**C**) NETosis-induced pathology: The excess production of NETs could exacerbate the different pathological defects associated with cerebral malaria (CM). During *Plasmodium falciparum* infection, the intravascular NETs activate the coagulation pathway, leading to microvascular occlusion and blood–brain barrier (BBB) disruption, contributing to the severity of malaria.

**Table 1 ijms-24-08975-t001:** Strategies elicited by different metazoan species to evade NETs-mediated killing.

Helminths (Metazoa)	Evasion Strategy	Factor/Molecule Involved	PMN Origin	Effect/Result	References
Cestodes *M. corti*	Inhibition of ROS-induced NETs by blocking (TRPM2) channel and calcium entry; downstream AMPK and autophagy.	Parasite excretory/secretory factors named parasitic ligands (PLs)	Mouse	Improved parasite survival; reduced systemic and local bacterial load.	[71]
Trematodes *S. japonicum*	Inhibition of PMA-induced NET formation by upregulating host IL-10 expression.	*S. japonicum* (SWAP) and *S. japonicum* (SEA)	Mouse	Parasitic survival	[63]
*F. hepatica*	Induction of weak NETosis; resolving the NETs; impairment of NETosis signaling pathways and active modulation of the (PMN) response.	Excretory/secretory Bovine (ES) molecules of *F. hepatica*		The parasitic stages (Eggs, metacercariae, and NEJ) escape the NETs-mediated killing.	[61]
Nematodes *T. spiralis*	Inhibition of PMA and microbe-induced release of NETs.	*T. spiralis* excretory/secretory (ES) antigens	Human	Establishment of the parasite inside the host; enhancing the parasitic penetration, migration, nutrition and survival.	[70]
Hookworms *Nb*	Degradation of the DNA backbone of NETs.	DNase II enzyme (Nb-DNase II)	Human; mouse	Prevention of larval damage and killing.	[64]
*Na*	Degradation of the DNA backbone of NETs.	DNase II enzyme	Human; mouse	Prevention of larval damage and killing.	[64]
*A. ceylanicum*	Inhibition of human (NE).	(AceKI-1); Soluble protein extracts and ES products	Human	Increased parasitic survival within the host intestine.	[73]
*A. duodenale*	Inhibition of human (NE).	AduTIL-1 *	Human	Increased parasitic survival within the host intestine.	[74]

PMN: Polymorphonuclear leukocytes; ROS: Reactive oxygen species; TRPM2: Transient Receptor Potential Melastatin 2; AMPK: Adenosine monophosphate-activated protein kinase; PLs: Parasitic ligands. PMA: Phorbol−12-myristate−13-acetate; SWAP: Soluble worm antigenic preparation; SEA: Soluble eggs antigen; NEJ: Newly excysted juvenile fluke; DNase: Deoxyribonuclease; NE: Neutrophil elastase. AceKI−1: A. ceylanicum Kunitz type inhibitor 1.* A novel serine protease inhibitor, named AduTIL−1, with two trypsin inhibitor-like (TIL cysteine-rich) domains from the adult *Ancylostoma duodenale.*

**Table 2 ijms-24-08975-t002:** Opposing roles of NETs in response to protozoan infections.

The Protozoan	The Stages Which Trigger (NETosis)	The (Jekyll and Hyde) Effect of (NETs) * (Protective/Harmful)	References
*Leishmania* spp. *L. amazonensis* *L. braziliensis* *L. donovani* *L. major* *L. infantum* *L. mexicana*	Promastigote LPG Amastigotes Promastigotes Promastigotes Promastigotes Promastigotes	NETs are lethal to the protozoan: Protective NETs are lethal to the protozoan: Protective The parasite evades killing using (LPG) The parasite evades killing using (LPG) The parasite evades killing using (3′NT/NU) NETs lack the killing potential: Harmful	[86] [87] [102] [103] [104] [99]
*Trypanosoma* spp. *T. cruzi (Y strain)* *T. brucei*, *T. evansi* *T. brucei*; *T. evansi*	Trypomastigotes; soluble Ag Trypomastigotes Trypomastigotes (live or dead)	NETs lack the killing potential: Harmful NETs reduce motility, yet lack the killing action The parasites evade killing using (TatD DNases)	[98] [96,97] [105]
*E. histolytica*	Trophozoites Trophozoites and EhLPPG Trophozoites	NETs are lethal to the protozoan: Protective NETs lack the killing potential: Harmful NETs promote inflammation and tissue damage: Harmful	[88] [100] [111]
*T. vaginalis*	Trophozoites Trophozoites and TvLPG/LG	NETs lack the killing potential: Harmful NETs are lethal to the protozoan: Protective	[101] [89]
*Plasmodium* spp. *P. falciparum*	Late-stage *P. falciparum*-iRBCs Haeme released from iRBCs *P. falciparum* ring stages	NETs contribute to pathophysiology of CM: Harmful NETs contribute to pathophysiology of CM: Harmful NETs contribute to the severity of human malaria: Harmful	[106] [107] [81]
*P. berghei*	*P. berghei*-iRBCs ^ND^	NETs contribute to malaria-associated respiratory distress: Harmful	[108]
*P. chabaudi*	*P. chabaudi*-iRBCs ^ND^	NETs contribute to liver damage: Harmful	[107]
*Apicomplexan* spp. *Eimeria* spp. *E. bovis* *E. arloingi* *E.ninakohlyakimovae*	Sporozoites Sporozoites and oocysts Sporozoites and oocysts	NETs prevent host cell invasion: Protective NETs lack the killing potential: Harmful NETs lack the killing potential: Harmful	[77] [92] [93]
*C. parvum*	Sporozoites and oocysts Sporozoites	NETs prevent host cell invasion: Protective NETs lack the killing potential: Harmful	[85] [91]
*T. gondii*	Tachyzoites ^a^ Tachyzoites ^b^	NETs are lethal to the protozoan and prevent host cell invasion: Protective NETs lack the killing potential: Harmful	[78] [90]
*N. caninum* *B. Besnoiti*	Tachyzoites	NETs prevent host cell invasion, yet lack the killing action	[95]
	Tachyzoites Tachyzoites	NETs prevent host cell invasion, yet lack the killing action NETs induce endothelial damage and cytotoxicity: Harmful	[94] [112]

* The Jekyll and Hyde effect of (NETs): refers to both positive and negative impacts of NETs during infections with different protozoa. LPG: lipophosphoglycan; 3′NT/NU: 3′-nucleotidase/nuclease enzyme; EhLPPG: *E. histolytica*-lipopeptidophosphoglycan; TvLPG/LG: *T. vaginalis*-lipophosphoglycan/lipoglycan. ND: Not determined; CM: cerebral malaria; iRBCs: infected red blood cells; ^a^: Tachyzoites incubated with human, mouse, and cattle neutrophils; ^b^: Tachyzoites incubated with sheep neutrophils.

## Data Availability

All data generated during this study are included within this article.

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
