# Peer review of "NETosis in Parasitic Infections: A Puzzle That Remains Unsolved"

_ijms, 2023, doi:10.3390/ijms24108975_

Round 1
Reviewer 1 Report
The authors reviewed the role of NETs with respect to parasitic infections. It appears as the authors are interested to present a discussion beyond the protective role or NETs in parasitic infection (protozoa and metazoan). However, it is not clear if their focus is protozoan or metazoan parasitic infection in the beginning. The article is informative but appears a little confusing at times since many different studies were mentioned without adequate details.
I would like to suggest a few things that the authors may want to consider in order for discussion presented to be more clear, informative and or complete.
Figures: Figures 1 and 3 appears busy and the animated character of the NETs don’t necessarily show how it interacts with the pathogen.
Line 16-21: It is not clear in the abstract if the authors want to focus on role of nets on metazoan or protozoa. It reads as they want to focus on metazoan but that is not necessarily evident in the paper.
Line 51: Under “2. Neutrophils and Neutrophil Extracellular Traps (NETs)”, it may be a good idea when introducing NETs and NETosis to indicate the role of these (what’s protective and harmful in general).
Line 63-65: Consider rewording the sentence-does the neutrophils nuclear DNA fibers only after exposure to bacteria? From what I understand, activated neutrophils release granule proteins and chromatin that form extracellular fibers. These extra cellular fibers bind Gram-positive and -negative bacteria. Naïve neutrophils don’t make these prominent extracellular structures. It would be good to mention here if pathogens can activate NET production in vitro/in vivo without stimulation.
Line 209-211: Does the NETs eventually kill the adult Onchocerca? Or is the “cloacking” used by the Onchocerca to hide from the eosinophils? Isn’t this section essentially about parasites evading the NET-mediated killing (the nest section)..so can they be together?
Author Response
"Please see the attachment."

Reviewer 2 Report
This is an excellent detailed review about "Netosis". Even tough it is focused on parasites, bacteria and viruses are mentioned. It might be worthwhile to add a few recent publications about these infectious agents to make "Netosis" more open to other fields for readers also interested in these fields.
Author Response
"Please see the attachment."

Reviewer 3 Report
This is a very thorough review examining the less well-reported area of the role of neutrophil extracellular traps (NETs) in parasite infection. It clearly explains the actions and effects of NETs in infections with a range of different parasites, and appears to be quite comprehensive in its examination of the literature. When finalised, this should be a useful addition to the literature.
Minor points to address:
Line 54: Neutrophils are not the most abundant leukocytes in all mammals, for example in mice they are much lower than in humans. This sentence should be modified accordingly.
Line 94: At the completion of the section on NETs and NETosis, it would be helpful to add in a brief statement about ‘vital’ NETosis and how, in some circumstances, neutrophils can remain active for some period of time following NETosis, rather than undergoing cell death (e.g. Yipp BG et al., Nat Med 2012, and PMID 24009232).
Abstract, + Lines 104, 107, 108 (and other places in the manuscript) – As far as I can tell, the term ‘metazoan’ applies to multicellular animals of all sorts. Perhaps it has another meaning in the parasite field but this could be clarified for a non-parasite specialist audience.
Line 148: The difference between ‘diffuse’ and ‘spread’ NETS should be defined.
Line 148-168: If NETs ‘entrap’ parasites but they don’t die, this observation is dependent on how long the observation lasts for. So it would be helpful if possible to explain what the ultimate outcome of entrapment is for the parasite. Does it get expelled, escape the entrapment or is there another outcome?
Line 208 – define Wolbachia
Line 319 – Just because neutrophil elastase is involved, that does not automatically mean that NETosis is involved. It could be just via degranulation.
Line 321 – perhaps just ‘mouse’ neutrophils, and why are they deemed ‘inflammatory’?
Line 428 – The use of the word ‘proven’ here is strange given that the subsequent papers provide evidence of the opposite. Perhaps this sentence could be recast in terms of evidence supporting one effect versus the other.
Author Response
Please see the attachment."

Reviewer 4 Report
I appreciate the authors for conceptualizing this review. However, I have to admit that the reading flow is unpleasant. It needs extensive English language editing.
Author Response
"Please see the attachment."

Round 2
Reviewer 4 Report
The reading pace is enjoyable. This review can be accepted for publication.
Author Response
Thank you so much for the reviewer's valuable comment and observation.
The article has undergone English language editing by MDPI. The text has been checked for correct use of grammar and common technical terms, and edited to a level suitable for reporting research in a scholarly journal.
" Please see the attached English editing certificate.
